# Multi-Criteria Comparative Analysis of Clean Hydrogen Production Scenarios

**Bartosz Ceran** 

Institute of Electric Power Engineering, Poznan University of Technology, Piotrowo 3A, 60-965 Poznan, Poland; bartosz.ceran@put.poznan.pl; Tel.: +48-61-665-2523

**Abstract:** Different hydrogen production scenarios need to be compared in regard to multiple, and often distinct aspects. It is well known that hydrogen production technologies based on environmentally-friendly renewable energy sources have higher values of the economic indicators than methods based on fossil fuels. Therefore, how should this decision criterion (environmental) prevail over the other types of decision criteria (technical and economic) to make a scenario where hydrogen production only uses renewable energy sources the most attractive option for a decision-maker? This article presents the results of a multi-variant comparative analysis of scenarios to annually produce one million tons of pure hydrogen (99.999%) via electrolysis in Poland. The compared variants were found to differ in terms of electricity sources feeding the electrolyzers. The research demonstrated that the scenario where hydrogen production uses energy from photovoltaics only becomes the best option for the environmental criterion weighting value at 61%. Taking the aging effect of photovoltaic installation (PV) panels and electrolyzers after 10 years of operation into account, the limit value of the environmental criterion rises to 63%. The carried out analyses may serve as the basis for the creation of systems supporting the development of clean and green hydrogen production technologies.

**Keywords:** photovoltaic-electrolysis; clean hydrogen; sustainable hydrogen production

---

## 1. Introduction

The Polish power industry is at a breakthrough point in the process of the structural transformation of the manufacturing sector. Currently, the production of one kWh of electricity in the national power system is burdened with an average $CO_2$ emission of 0.765 kg [1]. This is due to the fact that the main sources of electricity are still coal-fired power plants, accounting for 67% of the power installed in the system, of which 49% are fueled with hard coal and 18% are fueled by lignite. In the remainder, 33% are water power plants, 5% are industrial power plants, 6% are gas power plants, 6% are photovoltaic panels (PV), and 16% are wind power plants [2].

The transformation of the manufacturing sector from a central to a decentralized structure aims at increasing the share of renewable energy sources (RES) in the so-called distributed generation and at more efficiently reducing the carbon footprint and managing fossil fuels. Though coal will still be the main source of electricity, its consumption is projected to gradually decrease in favor of renewable energy sources [3,4].

Along with the development of the decentralized structure of electricity production, technologies of energy storage and the production of alternative fuels, e.g., hydrogen, are being developed [5]. The development of a decentralized structure based on stochastic sources is inextricably accompanied by the need to use energy storage technologies. Two examples are the German and Danish power systems, where the present structures of electricity generation are highly decentralized [6,7].

Hydrogen is expected to be one of the main energy carriers in the future, and thanks to the development of fuel cell technology, it will be widely used in both distributed generation [8] (including distributed cogeneration) and the transport sector [9].

The German government has set itself a goal of reducing $CO_2$ emissions by 55%, compared to 1990 levels, by 2030, and it is striving to achieve climate neutrality, i.e., net zero emissions by 2050. The German hydrogen strategy, which aims at the further decarbonization of the German economy, assumes increasing the capacity of electrolyzers producing pure hydrogen to 5 GW by 2030 and to 10 GW by 2040. The electrolysis process uses surplus renewable energy, including that produced by German off-shore wind farms. Berlin has adopted a target of increasing the installed capacity of offshore wind farms to 20 GW by 2030 and to 40 GW by 2040. Berlin plans to allocate a total of nine billion euros for the development of its national potential for the production of clean hydrogen [10].

A hydrogen infrastructure is being developed in Denmark, where the power system also has a decentralized structure. By 2030, Denmark plans to build a 1.3 GW electrolysis-based hydrogen production facility. The production facility is to annually produce 250,000 tons of hydrogen that will be used to power, among other things, hydrogen buses. As in the case of Germany, the hydrogen station will be powered by renewable energy produced by an offshore wind farm located near Bornholm. The project is to enable Denmark to meet its goal of reducing carbon dioxide emissions by 70% by 2030 [11].

Poland is one of the biggest hydrogen manufacturers in Europe, as it produces one million tons of this gas annually, which is 14% of hydrogen production in Europe. Hydrogen is mainly obtained in the steam reforming process, and it is the so-called grey hydrogen that is primarily used in the industry [12].

It is well-known that fuel cells with ion exchange polymer membranes need pure (99.999%) hydrogen to work. One method of hydrogen production to achieve such a purity is water electrolysis [13]. By supplying electrolyzers with energy from a power system, it is possible to produce pure hydrogen, but such hydrogen is not green because the energy is generated via the combustion of fossil fuels. Hydrogen is called green in the literature [14] if the energy supplied to electrolyzers is produced from renewable energy sources. Thus, the process of hydrogen production is not burdened with carbon dioxide emissions [15].

In [16], the authors showed that by producing hydrogen exclusively in the process of electrolysis, by using RES-powered electrolyzers, $CO_2$ emissions in the hydrogen production sector can be reduced by up to 75%. The high costs of the production process still remain a problem.

Therefore, issues related to sustainable hydrogen production should be considered in a way that takes many different and often conflicting criteria into account. Multivariate decision-making methods are now often used to solve energy problems [17].

The article presents a mathematical model designed for multivariate comparative analysis of electrolysis-based hydrogen production variants, taking the degree of utilizing green energy generated in PV installations to power electrolyzers into account.

The aim of the research was to determine the limit value for the weighting of the environmental criterion, for which the annual production of one million tons of hydrogen by PV–electrolyzer (El) systems is, under Polish conditions, the best option for the decision-maker.

The model allows for a comparative analysis of variants after *n* years of operation while taking component aging processes (PV and electrolyzer installations) into account, which is a novelty in this article.

This paper is organized as follows. Section 3 contains a description of a mathematical model designed to carry out a multi-variant comparative analysis of hydrogen production variants based on electrolysis. Section 4 presents the simulation results and examines the impact of the aging of PV systems and electrolyzers on the values of decision criteria. This section also includes a discussion of the results obtained. The conclusions of conducted research are included in Section 5.

## 2. Literature Review

Publications on hydrogen production in the process of electrolysis, with photovoltaics as the source of energy to power electrolyzers, have mainly focused on optimizing the system structure and, thus, minimizing hydrogen production costs.

These works concern low-temperature electrolyzers (alkaline and proton exchange membrane (PEM)). The properties of all types of electrolyzers (low and high temperature) can be found in [13,18]. In this article, the literature review focuses on the cooperation of PV with low-temperature electrolyzers.

In [19], the authors presented an example of a procedure to optimize a photovoltaic installation with an alkaline electrolyzer. The authors adopted the minimization of system construction costs as a function of the goal. In addition, the abandonment of an maximum power point tracker (MPPT) and a DC/DC converter was considered in order to avoid energy losses and improve system construction costs.

In turn, in [20], hydrogen production by a 10 kW electrolyzer was maximized. In order to optimize the structure of the system structure of an alkaline electrolyzer–PV installation, the authors used a genetic algorithm. By examining the influence of the system operating temperature on its efficiency, the authors minimized the generated excess power and energy transfer losses. The results of subsequent studies were presented in [21], where a wind turbine was added to the analyzed system. The authors presented a procedure to optimize the structure of a PV–wind–electrolyzer system using an imperialist competitive algorithm.

The authors of [22] presented a mathematical model implemented in the MATLAB/Simulink environment by means of which they adjusted the voltage and current characteristics of the electrolyzer to the power point of the maximum characteristic I = f(U) of the photovoltaic installation. The information obtained in this way allowed the authors to determine the number of cells with which the electrolyzer was to be built in order to make the production of hydrogen the most efficient under specific conditions.

In [23], the authors, by using a developed algorithm based on the multi-objective optimization technique, examined the influence of solar radiation, temperature, pressure, PV electrical efficiency, and the efficiency of an electrolyzer with a ion exchange polymer membrane on the value of voltage required for the electrolysis process.

The authors of [24] presented their own mathematical model that was designed to improve the efficiency of a PEM–PV electrolyzer system based on operating parameters such as voltage, current, temperature, and gas output pressure. As a function of the goal, the maximization of hydrogen production efficiency was adopted.

In [25], the authors carried out a cost analysis of the PV–El–FC system using the HOMER tool. The study was done by analyzing the impact of the dimensions of individual elements of the system, e.g., electrolyzer and PV installation, on the LCOE (levelized cost of electricity) cost index.

In turn, in [26], the authors examined the dynamic properties and limitations of the electrolyzer from the point of view of cooperation with a PV system in order to maximize hydrogen production.

In [27], the minimization of hydrogen production costs was adopted as a function of the goal of the optimization process. For this purpose, the authors determined the optimal ratio of peak power of the photovoltaic system to the nominal power of the electrolyzer.

Interesting research results were presented in [28], where the authors considered the impact of climatic conditions on the optimal size of a PV–electrolyzer system. The authors showed that climatic conditions had a significant impact on the system performance and should be taken into account in the process of optimizing PV–El system structures.

In general, research has been particularly focused on optimizing the structure of PV–electrolyzer systems and minimizing hydrogen production costs. It is justifiable because this technology of hydrogen production is one of the most expensive. Additionally, this solution has great potential to reduce $CO_2$ emissions. Bearing the development of electromobility and the progressive decentralization of the electricity production structure in mind, it should be forecast that the demand for pure hydrogen

will grow. At the same time, the European Union has imposed restrictive regulations regarding the reduction of $CO_2$ emissions [7,29].

Publications on the use of multi-variant decision-making methods to compare energy systems have also been the subject of research in recent years.

Multi-variant analyses can be used to compare small hybrid generation systems [30], as well as to assess transformation options for large energy systems [31].

In [32], the authors assessed energy and economic efficiency in the residential sector in Italy using a multi-criteria analysis. The analysis showed that in this particular climatic region, the best solution is to install solar collectors connected to heat pumps.

In [33], the authors, using a multi-variant decision-making method, compared the following hydrogen production technologies: using chemical energy for steam reforming, using electricity from nuclear power plants, using renewable energy from PV installations or wind turbines, and using energy from biomass. The best option in the comparison was found to be natural gas steam reforming. However, it cannot yield a hydrogen purity of 99.999%.

In [34], following a multivariate comparative analysis of hydrogen production methods in Brazil, the authors concluded that the best scenario in the comparison was the one where electrolyzers were powered by renewable energy sources. This was influenced by the structure of the power generation sector of the Brazilian electricity system, where about 70% of electricity comes from hydropower.

In [35], the authors compared technologies of hydrogen production via electrolysis with the use of renewable energy sources, applying an advanced method of multi-variant analysis based on the FRISCO formula. They came to the conclusion that the best alternative variant of hydrogen manufacture was the electrolysis of steam obtained by means of solar systems and electricity generated by PV systems.

Methods of multi-variant decision making are constantly being developed. For example, the authors of [36] proposed an innovative two-stage multi-criteria decision making method to select the most sustainable hydrogen production technology from the following variants: coal gasification with $CO_2$ capture, methane steam reforming, biomass gasification, biomass pyrolysis, wind-electrolysis, PV-electrolysis, hydropower-electrolysis, and nuclear-based high temperature electrolysis. The authors proved that the developed two-stage method could more accurately reflect a decision-maker's alternative preferences in regard to the selection of the best scenario from the various hydrogen production technologies.

However, the above-mentioned publications lacked long-term analyses, e.g., for a period of 10 years; by taking into account the decrease in the efficiency of PV and electrolyzer installations, such analyses would make it possible to precisely determine the cost of hydrogen production after *n* years of operation. In reviewed literature, optimization processes were carried out for the first year of operation of a pure green hydrogen production system. The previous studies did not take the reduction in the efficiency of the hydrogen production process associated with the decrease in equipment efficiency (i.e., PV panels and electrolyzers) into account. The authors of the above-mentioned works, presenting RES-El technology as the best way to produce hydrogen, were only guided by environmental criteria.

The issue of pure hydrogen production should be considered from a long-term perspective while taking not only the environmental criterion but also other criteria, e.g., technical, economic and logistic, into account.

## 3. Problem Description

The following variants of hydrogen production were compared:

- V1—100% of electrical energy intended to power the electrolyzers is supplied from the power system.
- V2—25% of the electricity for the power supply of the electrolyzers is supplied from a PV installation, and 75% of the demand is covered by the power system.

- V3—50% of the electricity intended to power the electrolyzers is supplied from a PV installation, and the remaining 50% of the demand is covered by the power system.
- V4—75% of the electricity intended to power the electrolyzers is supplied from the PV installation, and 25% of the demand is covered by the power system.
- V5—100% of electricity intended to power the electrolyzers is supplied from PV installations ($n$ 1 MW$_P$ PV farms).

A gradual increase of the share of energy from RES sources in the hydrogen production process reduces the $CO_2$ emissions associated with the production of this fuel. This issue is illustrated in Figure 1. The origin in the coordinate system represents the current situation, i.e., the production of one million tons of (grey) hydrogen per year for industrial purposes. The easiest way to produce pure hydrogen via electrolysis is to install an appropriate number of electrolyzers powered from the power system. Increasing the share of PV in hydrogen production takes time, and the greater the share of PV in the production process, the greater the time frame needed to complete the task.

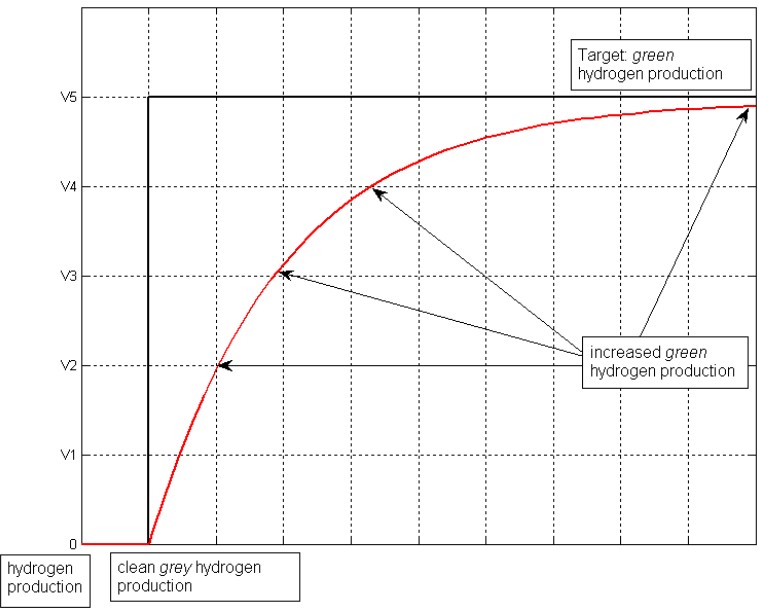

**Figure 1.** The way to variant V5: production of 100% pure and green hydrogen—own study.

According to the author, the achievement of the goal, variant V5, will be slow and spread over time. The time constant of this undertaking will depend on many factors, including technical (the development of hydrogen production technology, minimizing electrolyzer production costs, and minimizing the production costs of photovoltaic installations, structural changes in the manufacturing sector), economic, environmental (growing charges for carbon dioxide emissions), and political (appropriate support systems for the development of pro-environmental technologies). In [37], the authors demonstrated the relationship between the economic input and performance of a country and its positive impact on residents' perception of renewable energy based on hydrogen. The work showed that economic growth has a positive impact on the development of environmental awareness regarding the use of renewable energy and alternative fuels.

Water electrolysis with the use of electricity from a power system is a cheaper process of hydrogen production than using renewable energy sources today. The main reason for this is the low value of energy density produced by PV and the short time of using the power installed in this technology. Therefore, appropriate regulations will be required to place much more emphasis on the environmental aspects of pure hydrogen production. For this purpose, a weighting limit should be set on the environmental criterion

Guided by the environmental criterion alone, the best solution is to only use green energy so that $CO_2$ emissions are 0. However, due to decreasing equipment efficiency, hydrogen production will decline year by year. Following the economic criterion, production of pure (grey) hydrogen will be made cheaper. However, with each subsequent year, due to the growth of the unit energy required by the electrolyzer to produce 1 kg of hydrogen, total $CO_2$ emissions will increase. The problem should therefore be considered over a long-term period and in view of various decision criteria, including different aspects of hydrogen production.

The research presented in this article was aimed at establishing the relationship between the weighting of the environmental criterion and the weights of the technical and economic criteria in order to compare hydrogen production variants in the process of electrolysis using electrolyzers powered from a power system and PV installations. The compromise programming method, which is a multi-variant decision making method, was used in the study.

PV, the source of renewable energy in the considered variants, was selected for the following reasons. It is currently the most dynamically developing RES technology in Poland, and the power installed in PV is currently 1.7 GWp. The development of wind energy has been suspended by political decisions, while the potential of small hydropower plants is still unknown.

## 4. Model and Method Description

The block diagram of the hydrogen production process is shown in Figures 2–4.

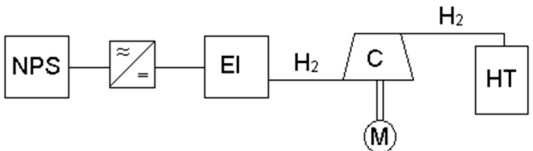

**Figure 2.** Flowchart: clean, grey hydrogen production system (El—electrolyzer; NPS—national power system; C—compressor; M—electric motor; and HT—hydrogen tank) [9].

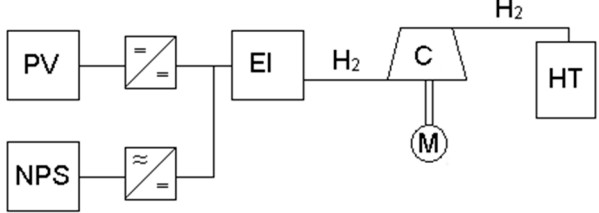

**Figure 3.** Flowchart: hydrogen production system (PV—photovoltaic installation) [9].

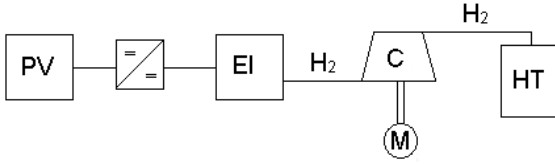

**Figure 4.** Flowchart: clean, green hydrogen production system [9].

The analysis assumed the use of 250 kW electrolyzers and 1 $MW_P$ photovoltaic farms built of 330 Wp PV panels. The value of annual operating time of the electrolyzer powered from the power network was assumed to be 7000 h.

To determine the demand for electricity needed to produce 1 million tons of pure hydrogen, Formula (1) was used:

$$A_{el} = m_{H_2} \cdot Q_{wH_2} \cdot \eta_{AC/DC}^{-1} \cdot \eta_{EL}^{-1} \cdot \eta_{comp}^{-1} \cdot 10^{-3} \tag{1}$$

where $A_{el}$ (MWh) is the total electricity demand from the power system, $m_{H2}$ (kg) is the mass of hydrogen produced, $Qw_{H2}$ is the calorific value of hydrogen (kWh/kg), $\eta_{AC/DC}$ is the efficiency of the converter, $\eta_{EL}$ is the efficiency of the electrolyzer (related to the calorific value of hydrogen), and $\eta_{comp}$ is the efficiency of the hydrogen compressor.

In order to determine the power required to be installed in PV systems to produce 1 million tons of pure hydrogen using green energy, the following relationship was used:

$$P_{PV} = m_{H_2} \cdot Qw_{H_2} \cdot \eta_{AC/DC}^{-1} \cdot \eta_{EL}^{-1} \cdot \eta_{comp}^{-1} \cdot T^{-1} \cdot 10^{-3} \tag{2}$$

where $P_{PV}$ (MW) is the total power installed in photovoltaic farms and T (h) is the time of using the power installed by the PV farm.

The efficiency of the hydrogen production process is influenced by the pressure value to which hydrogen is compressed (compressor operation), among other things. The energy required to compress one kilogram of hydrogen is shown as a function of pressure in Figure 5. For the purposes of this study, it was assumed that the produced hydrogen was compressed to 200 bar (20 MPa).

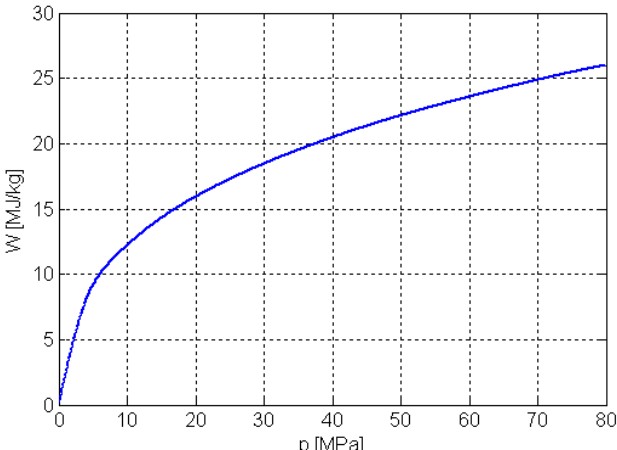

**Figure 5.** Dependence of work required to compress 1 kg of hydrogen as a function of final pressure [38].

Three benchmarking criteria were defined to compare the options considered.

The energy criterion is the unit power installed in electrolyzers for hydrogen production:

$$k_1 = P_{EL} \cdot m_{H_2}^{-1} [kW \cdot kg_{H_2}^{-1}] \tag{3}$$

This criterion has a major impact on the investment outlay in the individual variants because increasing the number of photovoltaic farms increases the number of electrolyzers, as per Equation (4) [39].

$$P_{EL} = max(P_{PV}) \tag{4}$$

This is related to the variability of electricity production by PV installations. To use 100% of energy produced by 1 MW of PV, four 250 kW electrolyzers must be installed (Figure 6).

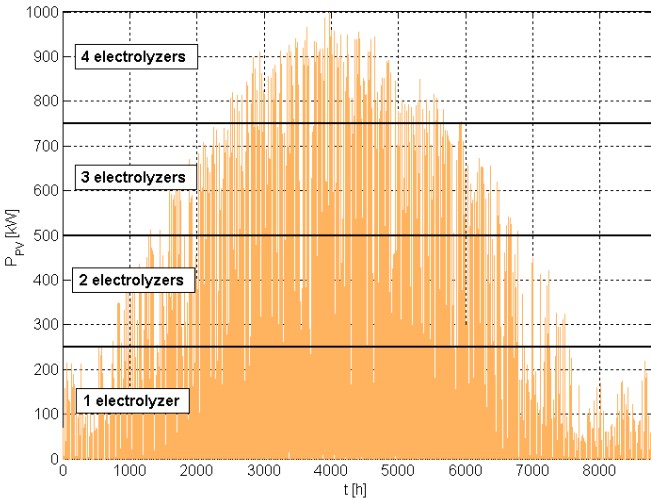

**Figure 6.** Impact of electricity production by PV installations on the number of working electrolyzers—own study.

The unit production cost of hydrogen defined by Equation (5) was adopted as the economic criterion:

$$k_2 = K_e \cdot m_{H_2}^{-1} [PLN \cdot kg_{H_2}^{-1}] \tag{5}$$

where $K_e$ represents the operating costs related to hydrogen production and $m_{H2}$ represents the mass of hydrogen produced.

Operating costs were defined in accordance with Equation (6):

$$K_e = k_a + k_{H_2O} + k_{O\&M} + k_E [PLN \cdot kg_{H_2}^{-1}] \tag{6}$$

where $k_a$ represents the depreciation costs, $k_{H_2O}$ represents the water purchase costs, $k_{O\&M}$ represents the operating and management costs, and $k_E$ represents the cost of energy purchase from the power system (in the case of variant V5, $k_E = 0$)

As the environmental criterion, the unit emission of carbon dioxide for the production of 1 kg of hydrogen was adopted.

$$k_3 = m_{CO_2} \cdot m_{H_2}^{-1} [kg_{CO_2} \cdot kg_{H_2}^{-1}] \tag{7}$$

where $m_{CO_2}$ is the mass of carbon dioxide emitted during the hydrogen production process.

The power of the PV installation can be represented by the following Formula (8):

$$P_{PV} = E \cdot S \cdot \eta_{PV} \tag{8}$$

where E represents the solar radiation intensity, S represents the PV installation surface, and $\eta_{PV}$ represents the PV panel efficiency.

An efficiency decrease in a photovoltaic installation is linked to the efficiency decrease of solar radiation conversion into electricity; see Formula (9):

$$\Delta P_{PV} = E \cdot S \cdot \Delta \eta_{PV} \tag{9}$$

where $\Delta P_{PV}$ represents the power drop caused by efficiency drop and $\Delta \eta_{PV}$ represents the PV panel efficiency drop.

Manufacturers of PV panels specify performance reduction characteristics for 20 years of operation, where the guaranteed value after 20 years is 80% of nominal performance. The current drop in performance is between 0.6% and 1% per year depending on the quality of the PV panels. The influence

of the decline of the PV panel's efficiency on the operational characteristic $P_{PV} = f(E)$ is shown in Figure 7.

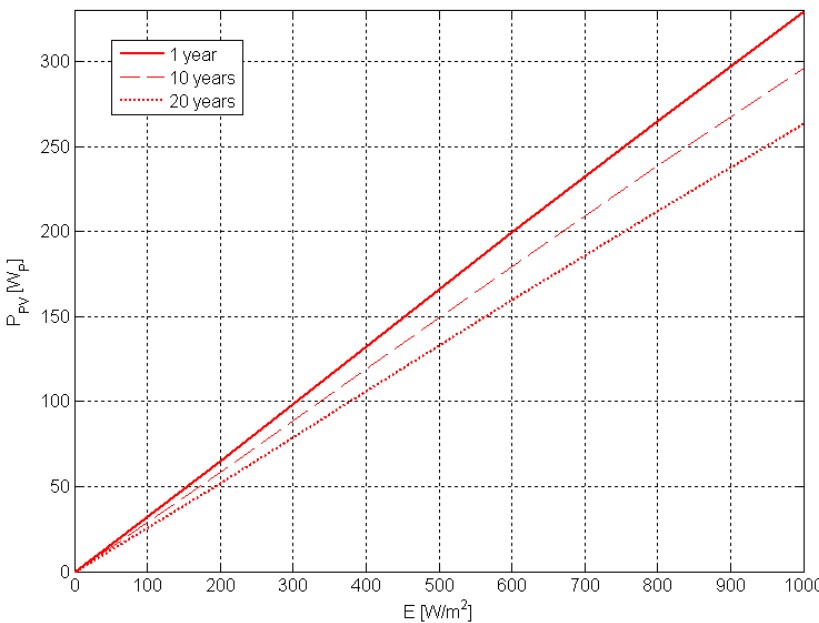

**Figure 7.** The influence of PV panel aging on $P_{PV} = f(E)$ [40].

The decrease in electrolyzer parameters was determined on the basis of the fuel cell stack aging model described in [41].

The efficiency (Low Heating Value (LHV)) of electricity conversion into fuel chemical energy by an electrolyzer can be presented with the use of Formula (10):

$$\eta_{EL} \; = \; n_{H_2} \cdot Q_{wH_2} \cdot P_{el}^{-1} \; = \; n_{H_2} \cdot Q_{wH_2} \cdot (U \cdot I)^{-1} \tag{10}$$

where $P_{el}$ represents the electrical power of the electrolyzer, U represents the voltage of the electrolyzer, I represents the current, $n_{H_2}$ represents the molar hydrogen flux, and $Q_{wH_2}$ represents the calorific value of hydrogen related to 1 mole of hydrogen.

The molar stream of produced hydrogen is proportional to the number of cells of which the electrolyzer is made, hence:

$$n_{H_2} \; = \; I \cdot n_{cells} \cdot (z \cdot F)^{-1} \tag{11}$$

For calculation purposes in electrochemistry, a quantity called thermoneutral potential is used, and this is defined according to Formula (12) [42]:

$$E_t^0 \; = \; -\Delta H_{H_2O(g)} \cdot (z \cdot F)^{-1} \tag{12}$$

where $E_t^0$ represents the thermoneutral potential (V), $\Delta H$ represents the standard enthalpy of water formation in the gaseous phase (kJ/mol), and index "0" means the standard conditions (T = 298 K, $p = 10^5$ Pa).

The standard enthalpy of water formation in the gaseous phase $\Delta H_{H2O(g)}$ energetically corresponds to the calorific value of hydrogen, assuming that the water is a product in the gaseous state; see Formula (13) [43]:

$$-\Delta H_{H_2O(g)} \; = \; Q_{wH_2} \tag{13}$$

By substituting Equations (11)–(13) for Equation (10), the formula for the efficiency of hydrogen production by the electrolyzer can be written as Equation (14):

$$\eta_{EL} = n_{cells} \cdot E_t \cdot U^{-1} \tag{14}$$

The average voltage value of an electrolyzer is defined as the ratio of the electrolyzer voltage to the number of cells (single cells) in the electrolyzer:

$$U_{av} = U \cdot n_{cells}^{-1} \tag{15}$$

After substituting Dependence (15) to Formula (14), the formula for electrolyzer efficiency can be presented as Correlation (16):

$$\eta_{EL} = E_t^0 \cdot U_{av}^{-1} \tag{16}$$

The change in electrolyzer efficiency is inversely proportional to the average voltage change.

$$\Delta\eta_{EL} = E_t^0 \cdot (\Delta U_{av})^{-1} \tag{17}$$

The Figure 8 shows the average voltage change in a 250 kW electrolyzer after 10 years of operation, based on [44].

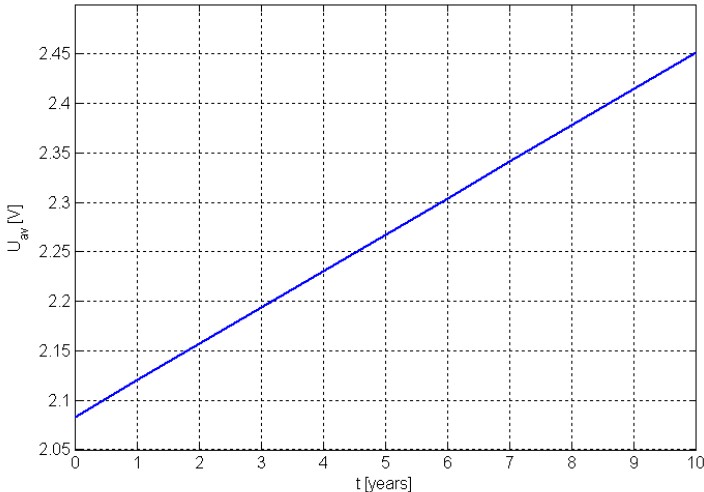

**Figure 8.** Average voltage growth in the electrolyzer as a function of years of operation—own study based on [44].

The multi-criteria analysis method used in the model allows one to compare variants according to defined criteria. The method consists of measuring the distance of a given variant from the so-called ideal point for which all normalized values of the criteria are 1. The idea of the method is graphically presented in Figure 9.

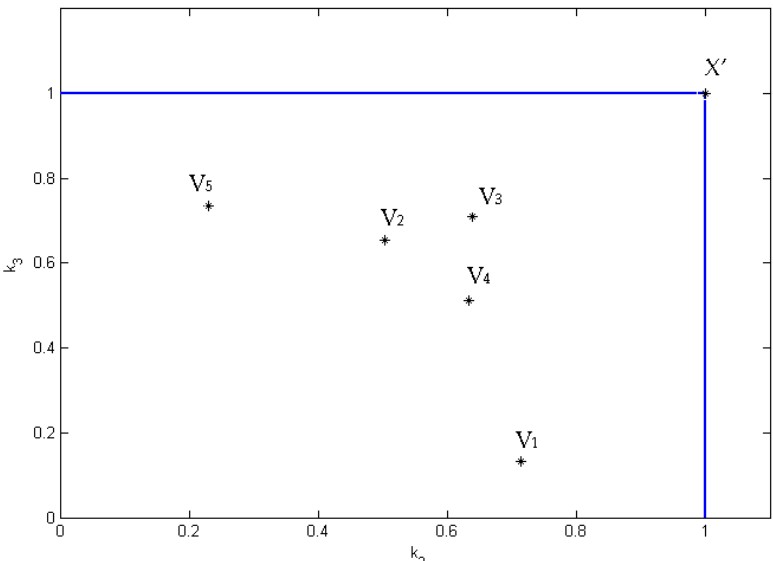

**Figure 9.** Graphical interpretation of target point X′ in the compromise programming method: k1 and k2—decision criteria; V1, V2, V3, V4, and V5—100% of electrical energy intended to power the electrolyzers is supplied from the power system, 25% of the electricity for the power supply of the electrolyzers is supplied from a PV installation, and 75% of the demand is covered by the power system, 50% of the electricity intended to power the electrolyzers is supplied from a PV installation, and the remaining 50% of the demand is covered by the power system, 75% of the electricity intended to power the electrolyzers is supplied from the PV installation, and 25% of the demand is covered by the power system, and 100% of electricity intended to power the electrolyzers is supplied from PV installations (*n* 1 MW$_P$ PV farms) scenarios, respectively [8].

The distance from the ideal point is measured by Equation (18):

$$L(V_n) \; = \; \sum_{m \, = \, 1}^{M} \omega_m^2 (x_m' - x_{nm}')^2 \tag{18}$$

where L(Vn) is distance of the nth variant from the ideal point, $x_{nm}'$ represents the normalized value of the assessment criterion, and $x_m'$ represents the mth coordinate of the ideal point.

The values of the decision criteria are normalized (reduced to a range of 0–1) by means of the following equation:

$$x_{nm}' \; = \; 1 - x_{nm} \cdot \left( \sum_{j \, = \, 1}^{m} x_{nm}^2 \right)^{-0.5} \tag{19}$$

where $x_{nm}$ represents the value of the assessment criterion.

The method was described in detail in [8].

## 5. Research Methodology

In order to determine the limit value of the environmental criterion for which the V5 scenario is considered the best option for hydrogen production, the author developed an algorithm implemented in MATLAB/Simulink environment on the basis of Equations (1)–(19).

The algorithm is graphically shown in Figure 10. The simulations allowed the author to determine the best variant of hydrogen production for a specific weight value of the environmental criterion.

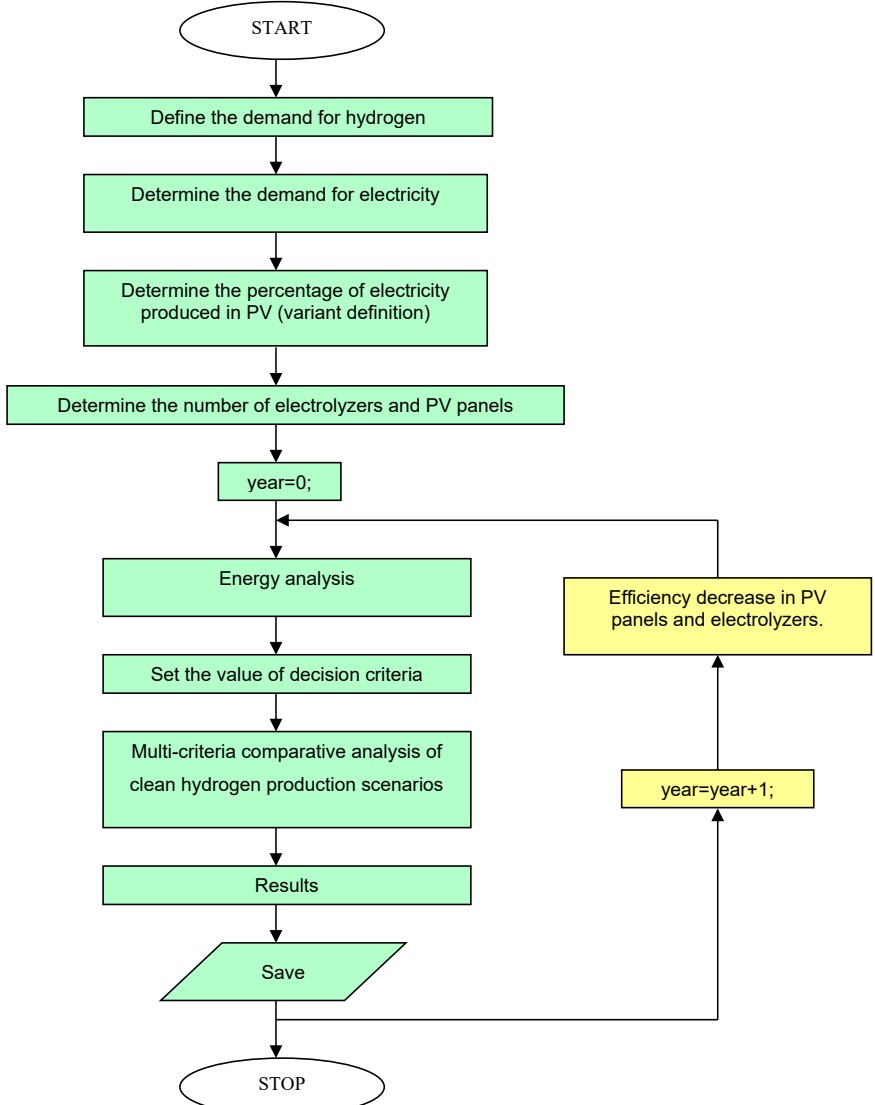

**Figure 10.** Block diagram of the model used to compare hydrogen production scenarios—own study.

In order to investigate the aging effect of PV panels and electrolyzers, the algorithm, based on the aging characteristics, updated the values of the decision criteria and determined the acceptable scenarios.

The simulations were performed with the following simplification assumptions:

- The quasi-determined state, in which the dynamic processes (switching the electrolyzers on and off) were not taken into account, was analyzed.
- During the one-year analysis period, random processes that may affect the energy yield, such as interruptions in the electricity supply from the power system (short circuit, power plant failures), were not taken into account.
- Energy supply interruptions from photovoltaic panels due to, e.g., failure of inverters, were not taken into account.

The results of the simulations and their discussion are presented in Section 6.

## 6. Results and Discussion

### 6.1. Results of the Energy Analysis

Following the energy analysis based on Equations (1)–(7), decision criteria values were determined for variants V1–V5. The results are presented in Table 1. Table 2 shows the values of the standardized decision criteria.

**Table 1.** Values of decision criteria—own study.

| Decision Criteria | Hydrogen Production Variants | | | | |
| --- | --- | --- | --- | --- | --- |
| | V1 | V2 | V3 | V4 | V5 |
| k1 (kW/kgH$_2$) | 0.0094 | 0.0235 | 0.0375 | 0.0516 | 0.0657 |
| k2 (PLN/kgH$_2$) | 41.030 | 64.500 | 87.960 | 111.43 | 136.21 |
| k3 (kgCO$_2$/kgH$_2$) | 50.240 | 37.680 | 25.120 | 12.560 | 0 |

**Table 2.** Standardized values of decision-making criteria—own study.

| Standardized Decision-Making Criteria | Hydrogen Production Variants | | | | |
| --- | --- | --- | --- | --- | --- |
| | V1 | V2 | V3 | V4 | V5 |
| k1 (min) | 0.901 | 0.753 | 0.605 | 0.457 | 0.309 |
| k2 (min) | 0.806 | 0.694 | 0.583 | 0.472 | 0.355 |
| k3 (min) | 0.270 | 0.452 | 0.635 | 0.817 | 1 |

Criterion $k_1$ grew in value as the share of PV installations grew. As mentioned above, this was related to the fact that four 250 kW electrolyzers have to be installed to use the electricity produced by a 1 MW$_p$ farm. Another factor that increased the value of criterion $k_1$ was the time of utilization of the installed capacity of a photovoltaic farm, which amounts to about 1000 h in Poland. This means that in order to produce the same amount of hydrogen per year as electrolyzers powered by the power system that operate at 7000 h/a of rated power, one need to build seven 1 MW$_p$ farms.

The $k_1$ value for the V1 variant was 0.0094 kW/kgH$_2$, which means that in order to produce 1 million tons of hydrogen, electrolyzers with a total capacity of 9400 MW would need to be installed, thus giving 37,600 electrolyzers of 250 kW. The amount of energy needed to feed the electrolyzers from the power system was, on the basis of Formula (1), found to be 65,668,505 MWh, and its production in the power grid system corresponded to the emission of 50.24 million tons of CO$_2$.

At the moment, it would not be a serious challenge to allocate such an amount of electricity to produce hydrogen. According to data from the power system operator, the installed capacity was 47 GW in December 2019, while the load in the system was between 22 and 24.4 GW [2]. As calculated on the basis of Formulas (5) and (6), the unit cost of hydrogen production amounts to PLN 41.03 (EUR 9.04).

The replacement of 25% of the energy used from the power grid with energy generated by PV (variant V2) was found to require the construction of 16,417 photovoltaic farms of 1 MW$_p$. The number of electrolyzers needed for such hydrogen production was 93,812, of which 28,144 were powered from the power system and 65,668 were powered from the PV installation. The value of criterion $k_1$ increased by 150% compared to the variant V1. The CO$_2$ emission index decreased by 12.56 kgCO$_2$/kgH$_2$. Cost increases linked to the costs of construction and operation of PV farms caused the value of the economic criterion $k_2$ to grow by 57.2%, as compared to variant V1.

The distribution of electricity production intended to supply electrolyzers in the proportion of 50/50 (variant V3), in regard the relation between black energy from the power system and green energy produced in PV systems, caused a 74.9% increase in the value of criterion $k_1$ compared to variant V1. This meant that the number of electrolyzers needed in this variant to produce 1 million tons of hydrogen per year is 150,100, of which 131,336 electrolyzers would have to cooperate with PV

installations and 18,763 electrolyzers would be powered from the power grid. The installed capacity of the PV farms was found to be 32,834 $MW_p$. The $CO_2$ emission index was 25.12 $kgCO_2$ /$kgH_2$ and was half the value of the index in variant V1. The emission reduction entailed a growth of unit operating costs, and the value of the economic criterion $k_2$ increased by 114.4% in relation to V1.

In the fourth variant (V4), with 75% of hydrogen production done by PV installations, the total installed power in the electrolyzers amounted to 51,600 MW, which translated into 206,400 electrolyzers, of which only 9400 were to be powered from the power system. The $k_3$ emission factor equaled 25% of its value in variant 1, while the cost of hydrogen production increased by as much as 171.6%.

As expected, V5 had the highest hydrogen production cost at 136.21 $PLN/kgH_2$, which was more than three times higher than in V1. The number of electrolyzers was 262,674. The total PV farm capacity was 65,668,505 $MW_p$ in this variant. For the Polish conditions, the value of the angle of inclination of photovoltaic modules offering the highest energy yield varied between 30° and 35°. An average area of 2 ha was found to be needed per 1 MW in Poland. The area required for the construction of such a number of 1 $MW_p$ PV installations was 131,337 ha, which represents 0.42% of Poland's territory.

When analyzing the standardized values of the decision-making criteria, it could be noticed that criterion $k_1$ favored variant V1, where the lowest number of electrolyzers was needed to produce the assumed quantity of hydrogen. The economic criterion $k_2$ also preferred variant V1. On the other hand, the environmental criterion $k_3$ preferred variant V5. The standard value of criterion $k_3$ for variant V5 was 1, which meant that the $CO_2$ emission factor was 0 in V5.

## 6.2. Effect of Aging on Decision-Making Criteria

By taking Correlations (9) and (17) into account, the values of decision-making criteria were determined for the next 10 years of operation while considering the increase in electricity demand for electrolyzers and the production efficiency decline in PV panels. The results of the analysis are presented in Table 3.

**Table 3.** The impact of efficiency decline in electrolyzers and PV farms on the values of decision-making criteria—own study.

| Criteria | V1 | | V2 | | V3 | | V4 | | V5 | |
|---|---|---|---|---|---|---|---|---|---|---|
| | Year_0 | Year_10 | Year_0 | Year_10 | Year_0 | Year_10 | Year_0 | Year_10 | Year_0 | Year_10 |
| $k_1$ | 0.0094 | 0.0094 | 0.0235 | 0.0249 | 0.0375 | 0.0423 | 0.0516 | 0.0622 | 0.0657 | 0.0849 |
| $\Delta k_1$ | 0 | | +0.0014 | | +0.0048 | | +0.0106 | | +0.0192 | |
| $k_2$ | 41.030 | 55.39 | 64.500 | 79.78 | 87.960 | 107.29 | 111.43 | 138.55 | 136.21 | 176.09 |
| $\Delta k_2$ | +14.36 | | +15.28 | | +19.33 | | +27.12 | | +39.88 | |
| $k_3$ | 50.240 | 59.1 | 37.680 | 44.33 | 25.120 | 29.55 | 12.560 | 14.78 | 0 | 0 |
| $\Delta k_3$ | +8.86 | | +6.65 | | +4.43 | | +2.22 | | 0 | |

When analyzing the results presented in Table 3, it can be seen that aging had the biggest impact on the value of economic criterion $k_2$. For variant V1, $k_2$ increased by 14.36 $PLN/kgH_2$. The growth of the unit cost of hydrogen production was affected by the increase in electricity consumption by electrolyzers and the price increase of electricity drawn from the power system. The study assumed a linear increase in electricity prices at the level of 2.7% per year [2]. A decreased electrolyzer efficiency was found to result in a growth of energy consumption for hydrogen production, which increased $CO_2$ emissions by 8.86 $kgCO_2/kgH_2$. The value of the decision criterion $k_1$ was not affected by the number of electrolyzers because it was constant throughout the whole considered period. The value of $k_1$ changed when the annual hydrogen production decreased. This only happened when hydrogen was produced in a PV–El system; therefore, the $k_1$ value did not change in V1 because the amount of produced hydrogen was constant (only the energy consumption for production purposes was higher).

When analyzing the growth of the $\Delta k_1$ value for subsequent variants, it could be seen that $k_1$ increased with each variant, and it was highest for variant V5, where its increase equaled +0.0192 kW/kg $H_2$. The value increase of the $k_1$ criterion was caused by the decline of hydrogen production by PV-powered electrolyzers. The aging effect impacted the amount of hydrogen produced. After 10 years of operation, the electricity produced by PV represented 90% of the energy produced in year one. The impact of the drop in panel capacity on the change in $k_1$ increased together with the growth of the installed PV farm capacity. In regard to variant V5, the declining PV production significantly reduced the annual hydrogen production, which amounted to 773,500,000 tons of $H_2$ after 10 years of operation.

Decreased hydrogen production, higher electricity prices, and increased energy consumption by electrolyzers powered from the power grid were found to result in an increased value of the economic criterion $k_2$. The biggest influence on the $k_2$ index value came from a decline in hydrogen production. In variant V5, hydrogen production cost increased by 39.88 PLN/kg $H_2$ after 10 years. For the sake of comparison, this increase amounted to +14,36 PLN/kgH$_2$ in variant V1.

The value of criterion $k_3$ was found to be influenced by the amount of electricity drawn from the power system to produce hydrogen. Therefore, the $k_3$ value was directly affected by the efficiency of the electrolyzer. Figures 11–13 present the value changes in decision criteria for variant V3.

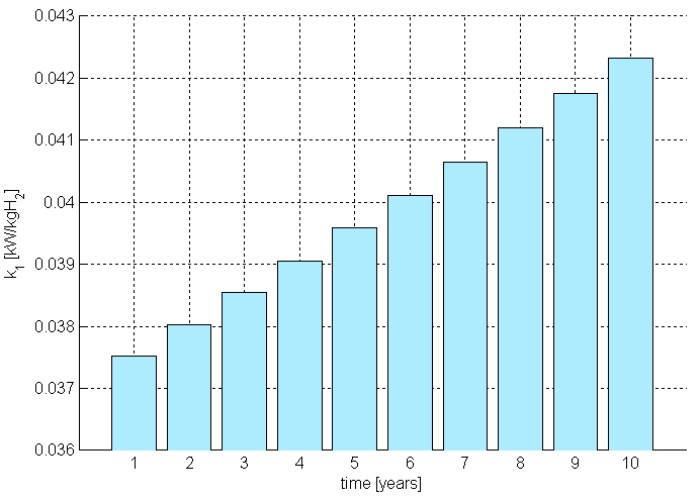

**Figure 11.** Change in the value of decision criterion $k_1$ for V3—own study.

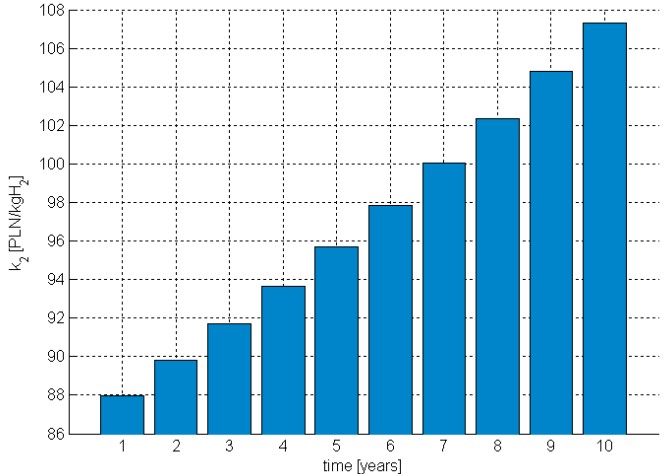

**Figure 12.** Change in the value of decision criterion $k_2$ for V3—own study.

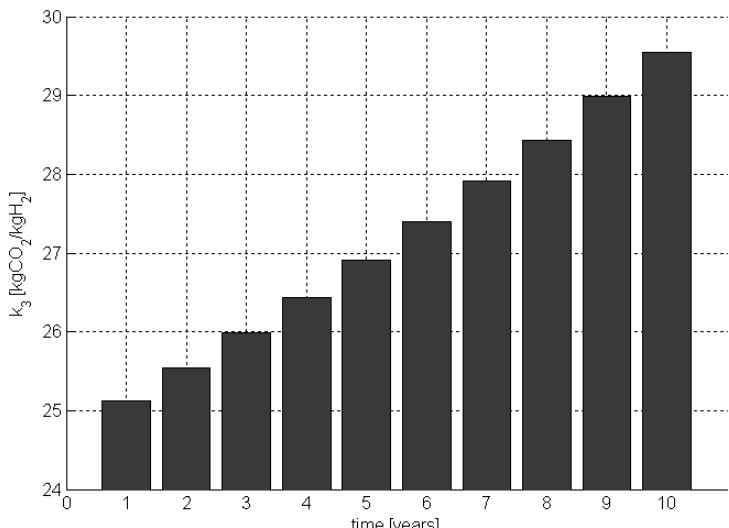

**Figure 13.** Change in the value of decision criterion $k_3$ for V3—own study.

*6.3. Limit Value of the Environmental Criterion Weighting*

The multi-variant analysis allowed for the selection of the best option in light of the adopted criteria and their weights. In this study, the aim was to answer the question of what the relationship between the weight of the environmental criterion and the weights of the other criteria should look like. As such, variant V5, where hydrogen production is the most expensive but free of $CO_2$ emissions, was considered the best. The research was carried out by increasing the environmental criterion weighting from 0 to 0.7 in steps of 0.01. For the adopted weighting value of criterion $k_3$, the weights of criteria $k_1$ and $k_2$ were equal and their sum added up to 100%. The simulations that were carried out made it possible to determine the weight value ranges for the environmental criterion, for which a given variant was indicated as the best. The results of the simulations are presented in Figure 14.

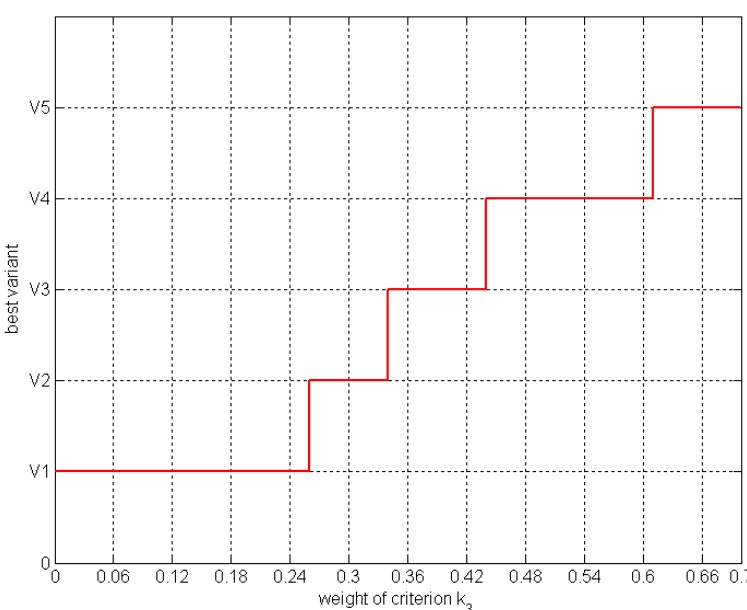

**Figure 14.** Influence of the weighting of criterion $k_3$ on the result of multi-variant analysis—own study.

Variant V1 was indicated as the best for the weight value of environmental criterion $k_3$ between 0% and 26%. Variant V2 was indicated as the best for the $k_3$ criterion weight falling in the range from 26% to 34%. It should be noted that within this weighting range, there was an even distribution of

weighting, i.e., in the ratio of 1:1:1. For a criterion weight of 1/3, the remaining criterion weight values also amounted to 1/3. Therefore, V2 was the variant that was indicated as the best for equal criterion weighting values, where 25% of the demand should be covered by a PV–electrolyzer system and the remaining 75% should be covered by electrolyzers in an electric power system.

Variant V3 was shown as the best for the environmental criterion weight of 34–44%. In this range, 50% of hydrogen production in PV–El was found to be an acceptable scenario. The range for the weighting of criterion $k_3$ for which V4 was the preferred option was between 44% and 61%.

In spite of exceeding the $k_3$ environmental criterion weight of 50%, V5 was still not considered to be the best option. The $k_3$ weighting limit for which V5 was indicated as the best was 61%. In order for scenario V5 to be considered the most preferred one, the environmental criterion weighting has to be over three times higher than the evenly distributed weighting of the other criteria.

By taking decreased equipment performance related to the aging processes into account, in accordance with Equations (9) and (17), simulations were run for the criteria values after 10 years of operation. The results are shown in Figure 15. The blue line was moved upwards on the vertical axis for better readability.

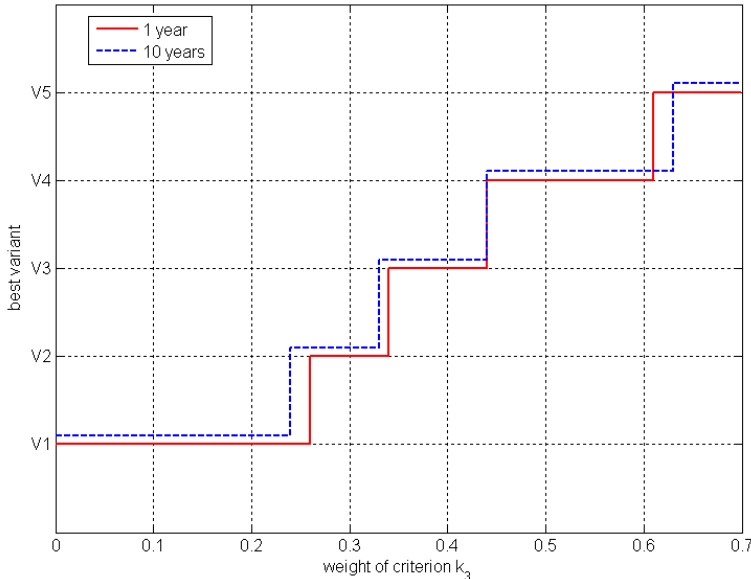

**Figure 15.** Effect of the aging of PV panels and electrolyzers on simulation results—own study.

When analyzing the impact of aging, it can be concluded that changes in criteria values have an ambiguous effect on criteria weighting limits. For the weighting range that favored variant V1, the weighting limit fell to 24%. On the other hand, the weighting limit for which V2 became the preferred option decreased to 33%. This was related to the decrease in electrolyzer efficiency, which entailed a growth of electricity consumption and $CO_2$ emissions. As a result, the introduction of PV already became more attractive for a lower weight value of the environmental criterion $k_3$.

The increase in $CO_2$ emissions had a smaller or zero impact on the distribution of the criteria weighting values for options IV and V. In these scenarios, the reduction of $CO_2$ emissions from hydrogen production was burdened with a large growth of hydrogen production unit costs. In option V5, the $CO_2$ effect was irrelevant as the value of the environmental criterion $k_3$ was 0. Therefore, the limit value of the criterion weight shifted to the right, increasing by 2 p.p. V5 became an attractive solution for the value of the environmental criterion weight of 63%. The remaining criteria then had a weighting of no more than 18.5%.

### 7. Conclusions

The article presents the results of a multivariate comparative analysis of hydrogen production scenarios based on electrolysis. In order to fully describe all aspects of the production of pure green hydrogen via electrolysis using renewable energy sources, such analyses had to be run in a multi-criteria format that took the effect of equipment efficiency decline associated with long-term operation into account.

The simulations showed that with an even distribution of weights over all the three decision criteria, the best variant was V2, where 25% of hydrogen production was done using PV installations. The simulation studies showed that the weighting of the environmental criterion must be at least 63% in order for the variant where hydrogen production via electrolysis is only based on PV as the source of electricity to be the best variant over 10 years of operation. The decreased operating efficiency of electrolyzers and PV panels after years of operation was found to result in a decrease in the assumed hydrogen production and an increase in the environmental criterion weight limit by two percentage points. The larger the scale of PV production, the greater the impact of the decrease in equipment efficiency on the production results.

The research showed that the reduction of $CO_2$ emissions in the process of hydrogen production via electrolysis by replacing electricity from the power grid with energy from PV installations increases the unit cost of hydrogen production by 213%. Such a high cost of hydrogen production renders it impossible to implement this variant without appropriate financial support mechanisms from the Polish government.

Therefore, the results of the conducted research and analyses may constitute the basis for the creation of the Polish energy policy in the period of power system transformation from a central structure based on coal to a decentralized structure through an appropriate support mechanism. The government, as a decision-maker, must indicate that the environmental criterion is becoming a priority here.

Further studies will focus on updating the proposed model. The author plans to take random phenomena impacting the annual effect of hydrogen production into account by introducing further decision criteria, including indicators of equipment operation reliability and those of energy supply continuity from the power system.

**Author Contributions:** Conceptualization, B.C.; methodology, B.C.; software, B.C.; validation, B.C.; formal analysis, B.C.; investigation, B.C.; resources, B.C.; data curation, B.C.; writing—original draft preparation, B.C.; writing—review and editing, B.C.; visualization, B.C.; supervision, B.C.; project administration, B.C.; funding acquisition, B.C. The author has read and agreed to the published version of the manuscript.

**Funding:** This research was funded by the Ministry of Science and Higher Education, grant number 0711/SBAD/4412.

**Conflicts of Interest:** The author declares no conflict of interest.

### Nomenclatures

*Indexes*

| | |
|---|---|
| $m$ | number of options in the multi-optional analysis |
| $n$ | number of criteria in a multi-optional analysis |

*Parameters*

| | |
|---|---|
| $\omega_m$ | weighting of the mth decision criterion |
| $k_a$ | amortization charges |
| $k_E$ | costs of purchasing electricity from the power grid |
| $k_{H2O}$ | cost of purchasing water |
| $k_{O\&M}$ | operating and management costs |
| $m_{CO2}$ | mass of carbon dioxide emitted during the hydrogen production process |

| | |
|---|---|
| $m_{H2}$ | mass of hydrogen produced |
| $n_{cell}$ | number of cells forming the electrolyzer |
| $n_{H2}$ | hydrogen molar stream |
| $x'_{nm}$ | normalized value of an assessment criterion |
| $x_m$ | mth coordinate of the ideal point |
| $x_{nm}$ | value of the assessment criterion |
| $z$ | number of electrons required to release the particle |
| $\eta_{AC/DC}$ | AC/DC converter efficiency |
| $\eta_{comp}$ | compressor efficiency |
| $\eta_{DC/DC}$ | DC/DC converter efficiency |
| $\eta_{El}$ | efficiency of the electrolyzer |
| $\Delta H$ | standard enthalpy of water creation |
| $A_{el}$ | total demand for electricity from the power system |
| $E$ | solar irradiance |
| $E_t$ | thermoneutral potential |
| $F$ | Faraday constant |
| $I$ | current flow of the electrolyzer |
| $K_e$ | operating costs related to hydrogen production |
| $L(Vn)$ | distance of the nth variant from the ideal point |
| $P_{el}$ | nominal power of electrolyzers |
| PLN | currency in Poland |
| $P_{PV}$ | peak power of PV panels |
| $Q_{wH2}$ | hydrogen calorific value |
| $S$ | total PV installation area |
| $T$ | the time of using the installed PV capacity |
| $U$ | nominal voltage of the electrolyzer |
| $U_{av}$ | average voltage value of one cell |
| *Variables* | |
| $k_1$ | energy criterion |
| $k_2$ | unit cost of hydrogen production |
| $k_3$ | specific $CO_2$ emission factor |

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
