# Peer review of "Multi-Criteria Comparative Analysis of Clean Hydrogen Production Scenarios"

_energies, doi:10.3390/en13164180_

Round 1

Reviewer 1 Report

In the manuscript, the author presented the results of multi-variant comparative analysis of annual production scenarios of 1 million tons of pure hydrogen in the electrolysis process in Poland.

The topic of the paper is interesting as well as the potential academic contribution of the work, but the author should improve the work according to the following indications.

1. The abstract should be rewritten according to the standard.

2. In the introduction, the authors should discuss international situation, regulations, and approaches, and should motivate the research to be of high interest for a even broader group of addressees.

3. Literature review is absent, and should be addressed. Moreover, the following study should be considered: https://econjournals.com/index.php/ijeep/article/view/6255

4. The author should discuss how the results can be interpreted in perspective of previous studies and of the working hypotheses.

5. Policy implications, limitations of the study and future research directions should be addressed.

6. The conclusions should be rewritten according to the standard.

7. Tables and figures should report the sources.

8. Extensive editing of English language and style required.

Author Response

Reviewer 1

The topic of the paper is interesting as well as the potential academic contribution of the work, but the author should improve the work according to the following indications.

Response

Thank you very much to the reviewer for their time and constructive comments that made the article clearer for the readers.

1. The abstract should be rewritten according to the standard.

Thank you for this comment. Abstract, as suggested by the reviewer, has been corrected.

I marked it in the manuscript.

“Different hydrogen production scenarios need to be compared with regard to multiple, and often distinct aspects. It is well known that hydrogen production technologies based on environment-friendly renewable energy sources, will have higher values of the economic indicators than methods based on fossil fuels. How, therefore, should this decision criterion () prevail over the other types of decision criteria (technical, economic) to make a scenario where hydrogen production only uses renewable energy sources the most attractive option for the decision-maker? This article presents the results of multi-variant comparative analysis of scenarios to produce annually 1 million tons of pure hydrogen (99.999%) via electrolysis in Poland. The compared variants differ in terms of electricity sources feeding the electrolyzers. The research carried out demonstrated that the scenario where hydrogen production uses energy from photovoltaics only becomes the best option for the environmental criterion weighting value at 61%. Taking into account the ageing effect of PV panels and electrolyzers after 10 years of operation, the limit value of the environmental criterion rises to 63%. The analyses which were carried out may serve as the basis for the creation of systems supporting the development of clean and green hydrogen production technologies.”

2. In the introduction, the authors should discuss international situations, regulations, and approaches, and should motivate the research to be of high interest for an even broader group of addresses.

Thank you for your comment. In accordance with the thematic requirements of the special issue, the author described the situation in Germany and Denmark, i.e. countries implementing the decentralization of electricity production with the development of hydrogen infrastructure.

I marked it in the manuscript.

“The German government has set itself a goal of reducing CO2 emissions by 55% by 2030 compared to 1990 levels, and it is striving to achieve climate neutrality, i.e. net zero emissions by 2050. The German hydrogen strategy, which aims at further decarbonisation of the German economy, assumes increasing the capacity of electrolyzers producing pure hydrogen to 5 GW by 2030 and to 10 GW by 2040. The electrolysis process is to use surplus renewable energy, including that produced by German off-shore wind farms. Berlin has adopted a target of increasing the installed capacity of offshore wind farms to 20 GW by 2030 and to 40 GW by 2040. Berlin plans to allocate a total of 9 billion euros for the development of its national potential for the production of clean hydrogen [10].

A hydrogen infrastructure is being developed in Denmark, where the power system also has a decentralized structure. By 2030, The Danes plan to build a 1.3 GW electrolysis-based hydrogen production facility by the year 2030. The production facility is to produce annually 250,000 tonnes of hydrogen, which will be used to power, among other things, hydrogen buses. As in the case of Germany, the hydrogen station will be powered by renewable energy produced by an offshore wind farm located near Bornholm. The project is to enable Denmark to meet its goal of reducing carbon dioxide emissions by 70% by 2030 [11].”

3. Literature review is absent, and should be addressed. Moreover, the following study should be considered https//ecojurnals.com/index.php.ijeep/article/view/6255

Thank you for your comment. Literature has been expanded to include the following articles:

    • Siksnelyte, I.; Zavadskas, E.K.; Streimikiene, D.; Sharma, D. An Overview of Multi-Criteria Decision-Making Methods in Dealing with Sustainable Energy Development Issues. Energies 2018, 11(10), 2754
    • Ceran, B.; Kaźmierczak, Ł. The use of Multicriteria Comparative Analysis to the Selection of the Standalone Hybrid Power Generation System Based on Renewable Energy Source. 15th International Conference on the European Energy Market (EEM), 2018, Łódź, Poland, 1-5.
    • Volkarta, K.; Weidmann, N.; Bauer, C.; Hirschberg, S. Multi-criteria decision analysis of energy system transformation pathways: A case study for Switzerland. Energy Policy, 2017, 106, 155-168.
    • Campisi, D., Gitto, S., Morea, D. An Evaluation of Energy and Economic Efficiency in Residential Buildings Sector: A Multi-criteria Analisys on an Italian Case Study. International Journal of Energy Economics and Policy, 2018, 8(3), 185-196.
    • Afgan, N.H.; Veziroglu, A.; Carvalho M.G. Multi-criteria evaluation of hydrogen system options. Int J Hydrogen Energ 2007, 32, 3183-3193.
    • Tapia L.C.F., Vigouroux R.Z.; Silveira J.L. Sustainability Assessment of Hydrogen Production Techniques in Brazil: A Multi-criteria Analysis, Print ISBN: 978-3-319-41614-4, Electronic ISBN: 978-3-319-41616-8
    • Badea, G.; Naghiu, G. S.; Felseghi, R. -A.; Raboaca, S.; Aschilean, I.; Giurca, I. Multi-criteria analysis on how to select solar radiation hydrogen production system. AIP Conference Proceedings, 2015, 1700, 050011.
    • Ren, X.; Shimin W.L.; Donga, D.L. Sustainability assessment and decision making of hydrogen production technologies: A novel two-stage multi-criteria decision making method. Int J Hydrogen Energ 2020, Available online

4. The author should discuss how the results can be interpreted in perspective of previous studies and of the working hypotheses

The results of the analyzes can be used to build an appropriate energy policy and organize, for example, development support systems, following the development and support of renewable energy production technologies or cogeneration technologies.

The author has placed his considerations in Conclusions in point 6.

5. Policy implications, limitations of the study and future research directions should be addressed

The author put it in “Conclusios” and in the “research methodology” chapter:

“research methodology”

The simulations are performed with the following simplification assumptions:

  • quasi-determined state is analyzed, in which the dynamic processes (switching the electrolyzers on and off ) are not taken into account,
  • during the one-year analysis period, random processes that may affect the energy yield, such as interruptions in the electricity supply from the power system (short circuit, power plant failures) are not taken into account,
  • energy supply interruptions from photovoltaic panels due to, e.g. failure of inverters, are not taken into account.

The results of the simulations and their discussion are presented in chapter 6.

“Conclusion”

I added conclusions in the next point.

6. The conclusions should be written according to the standard.

Thank you very much for this comment. I agree with you. I took into account your comments revising my manuscript. I hope that this time it will comply with the adopted requirements.

The article presents the results of a multivariate comparative analysis of hydrogen production scenarios based on electrolysis. In order to fully describe all aspects of the production of pure green hydrogen via electrolysis using renewable energy sources, such analyses must be run in a multi-criteria format, , taking into account the effect of equipment efficiency decline associated with long-term operation.

The simulations carried out have shown that with an even distribution of weights over all the three decision criteria, the best variant is V2, where 25% of hydrogen production is done using PV installations. The simulation studies have shown that the weighting of the environmental criterion must be at least 63% in order for the variant where hydrogen production via electrolysis is only based on PV as the source of electricity, to be the best variant over 10 years of operation. Decreased operating efficiency of electrolyzers and PV panels after years of operation results in a decrease in the assumed hydrogen production and an increase in the environmental criterion weight limit by 2 percentage points. The larger the scale of PV production, the greater the impact of the decrease in equipment efficiency on the production results.

The research has shown that the reduction of CO2 emissions in the process of hydrogen production via electrolysis by replacing electricity from the power grid with energy from PV installations, increases the unit cost of hydrogen production by 213%. Such a high cost of hydrogen production renders it impossible to implement this variant without appropriate financial support mechanisms from the Polish government.

Therefore, the results of the conducted research and analyses may constitute the basis for the creation of the Polish Energy Policy in the period of power system transformation from a central structure based on coal to a decentralized structure, through an appropriate support mechanism. The government, as a decision-maker, must indicate that the environmental criterion is becoming a priority here.

Further studies will focus on updating the proposed model. The author plans to take into account random phenomena impacting the annual effect of hydrogen production by introducing further decision criteria, including indicators of equipment operation reliability and those of energy supply continuity from the power system.”

7. Tables and figures should report the sources

Thank you for the comment.

Figure 1 is my authorship, it has not been published anywhere before, so I added my own development under the picture.

Figure 6 is my authorship, it has not been published anywhere before, so I added my own development under the picture.

Tables 1, 2 and 3 present the results of my author's simulation research.

In other cases, I added the sources.

 8. Extensive editing of English language and style required

Thank you for this remark. I am really sorry, that my language skills interrupted the reception of my work. According to your suggestions, I used a professional English editing service during revision.

Reviewer 2 Report

This is a very interesting manuscript on the future grey and green hydrogen production options and the multi-variant analysis with a focus on renewable penetration (PV) in the input parameters of the process.

The manuscript would be publishable to the journal, following some kind of restructuring in order to reveal the novelty and the originality. A clear methodology approach that is used is needed in a brief paragraph.

Especially:

line 76: ...analised". Proofread language is needed for the whole manuscript.
line 128-131: Define the aim of the paper in a clear way.

There is no clear the operation of the weighting criterion in the models V1-V5.

Model and flows assumptions have to further described and deeper analyzed. The differences in the production of hydrogen have to be captured in the modeling process. For example, the compression to 200 bar is one of them but a detailed analysis of the model is suggested. The limitations and boundaries of the flow charts have to be defined.

A graphic presentation to determine the flows of inputs and outputs of the model is suggested.
Scenarios are missing the description and the assumptions made.
Νomencalture needs checking again because some parameters are missing (equation 18, ω). What is the meaning of the operator? Define it in the methodology brief analysis.

The manuscript needs restructuring and submission again.

Author Response

Reviewer 2

This is a very interesting manuscript on the future grey and green hydrogen production options and the multi-variant analysis with focus on renewable penetration (PV) in the input parameters of the process.

The manuscript would be publishable to the journal, following some kind of restructuring in order to reveal the novelty and the originality. A clear methodology approach that is sued is needed in a brief paragraph.

Response

Thank you for your time dedicated for reviewing my manuscript. Below I provide the detailed response to your comments.

  • line 76 ...analised"/ Proofread language is needed for the whole manuscript.

Thank you for this remark. I am really sorry, that my language skills interrupted the reception of my work. According to your suggestions, I used a professional English editing service during revision.

  • line 128-131: define the aim of the paper in a clear way

Thank you for this comment. As suggested by the reviewer, I specifically defined the purpose of my research. I put in introduction.

“The aim of the research was to determine the limit value for the weighting of the environmental criterion for which the annual production of 1 million tonnes of hydrogen by PV-El systems only is, under Polish conditions, the best option for the decision-maker.”

  • there is no clear the operation of the weighting criterion in the model V1-V5

Thank you for this comment.

It was in the first version of the manuscript.

Line 437-439

“The research was carried out by increasing the weighting of the environmental criterion from 0 to 0.7 with a step of 0.01. For the adopted value of the weighting of criterion k3, the weights of criteria k1 and k2 were equal and their sum was a complement to 100%.”

And in revised version:

lines 501-503

“The research was carried out by increasing the environmental criterion weighting from 0 to 0.7 in steps of 0.01. For the adopted weighting value of criterion k3, the weights of criteria k1 and k2 were equal and their sum was the complement to 100%.”

  • Model and flows assumptions have to further described and deeper analyzed. The differences in the production of hydrogen have to be captured in the modeling process. For example, the compression to 200 bar is one of them but detailed analysis of the model is suggested. The limitations and boundarie of the flow charts have to be defined.

Thank you for your comments, limitations and boundarie made in an additional section.

“research methodology”

The simulations are performed with the following simplifying assumptions:

  • quasi-determined state is analyzed - in which the dynamic processes (switching on, off electrolysers) are not taken into account,
  • during the one-year period of the analysis, random processes that may affect the energy yield, such as: interruptions in the supply of electricity from the power system (short circuit, power plant failures) are not taken into account
  • interruptions in the energy supply from photovoltaic panels due to e.g. failure of inverters.

The results of the simulations and their discussion are presented in chapter 6.

  • A graphic presentation to determine the flows of inputs and outputs of the model is suggested. Scenarios are missing the description and the assumptions made. Nomenclature needs checking again because some parameters are missing (equation 18 ω). What is the meaning ot the operator? Define it in the methodology brief analysis.

A block diagram representing the calculation algorithm has been added. The assumptions apply to the entire model. The scenarios differ only in the flow of energy intended for the production of hydrogen (to power the electrolysers).

Figure 10. Block diagram of the model used to compare scenarios of hydrogen production – own dewelopment. (doc file)

Thank you, omega means the weight of the criterion and is in the nomenclature as the first item in parameters.

Line 519 – first version of the manuscript

ωm - weighting of the m-th decision criterion

line 596 - revised version

Reviewer 3 Report

  1. Every scientific article must contain, in different forms, the following section: “Introduction”, “Literature review”, “Research methodology”, “Calculation & Results”, “Conclusions”. From this point of view, the author respects just partly this structure (for instance, I can’t see the “Literature review section”; furthermore, the structure is quite unbalanced – the “Introduction” section is too large, comparing with other section). Also, I don’t see the “Research methodology” section (the author must present the main step of the methodology from theoretically point of view, with a short description for each of them). Therefore, I suggest the author to reshape the paper, according to the structure presented above.
  2. All the figures and tables in the paper must present the source (just as an example, in this form of the paper I can’t see the source for figure 1, figure 2, table 1 etc.).
  3. “Conclusion” section is too short, an improvement is mandatory. Furthermore, this section looks like a scheme, with those bullets.
  4. In direct connection with point 3, the author does not emphasize the practical implications of this research study. It would be very interesting to see if the main findings of the research would be useful to be implemented in practice. Or maybe if the findings can explain some phenomena within the Poland economic / social / energetic reality.
  5. Also, I don’t see the limitations of the research study.
  6. “References” section must be improved with other significant studies in this scientific area.

Author Response

Reviewer 3

Response

Thank you for your time dedicated for reviewing my manuscript. Below I provide the detailed response to your comments.

  1. Every scientific article must contain, in different forms, the following section "Introduction" "Literature review ", "research methodology", "calculations and results", ''conclusions" From this point of view, the autor respects just partly this structure (for instance, I cant see the "literature reviev section", furthermore, the structure is quit unbalanced - the "Introduction" section is to large, comparing with other section). Also, I don't see the "research methodology" section (the author must present the main step of the methodology from theoretically point of view, with a short description for each of them). therefore, I suggest the author to reshape the paper, according to the structure presented above.

The structure of the article has been changed according to the reviewer's recommendations.

New article structure:

"Introduction"

"Literature review",

“Problem description”,

“Model and Method Description”,

"Research Methodology",

„Results and discussion “ ,

'' Conclusions”.

  1. All the figures and tables in the paper must present the source (just as an example, in this form of the paper I can't see the source for figure 1, figure 2, table 1 ect)

Thank you for the comment.

Figure 1 is my authorship, it has not been published anywhere before, so I added my own development under the picture.

Figure 6 is my authorship, it has not been published anywhere before, so I added my own development under the picture.

Tables 1, 2 and 3 present the results of my author's simulation research.

In other cases, I added the sources.

  1. "conclusions" section is too short an improvement is mandatory Furthermore, this section looks like a scheme with those bullets

Thank you very much for this comment. I agree with you. I took into account your comments revising my manuscript. I hope that this time it will comply with the adopted requirements.

“The article presents the results of a multivariate comparative analysis of hydrogen production scenarios based on electrolysis. In order to fully describe all aspects of the production of pure green hydrogen via electrolysis using renewable energy sources, such analyses must be run in a multi-criteria format, , taking into account the effect of equipment efficiency decline associated with long-term operation.

The simulations carried out have shown that with an even distribution of weights over all the three decision criteria, the best variant is V2, where 25% of hydrogen production is done using PV installations. The simulation studies have shown that the weighting of the environmental criterion must be at least 63% in order for the variant where hydrogen production via electrolysis is only based on PV as the source of electricity, to be the best variant over 10 years of operation. Decreased operating efficiency of electrolyzers and PV panels after years of operation results in a decrease in the assumed hydrogen production and an increase in the environmental criterion weight limit by 2 percentage points. The larger the scale of PV production, the greater the impact of the decrease in equipment efficiency on the production results.

The research has shown that the reduction of CO2 emissions in the process of hydrogen production via electrolysis by replacing electricity from the power grid with energy from PV installations, increases the unit cost of hydrogen production by 213%. Such a high cost of hydrogen production renders it impossible to implement this variant without appropriate financial support mechanisms from the Polish government.

Therefore, the results of the conducted research and analyses may constitute the basis for the creation of the Polish Energy Policy in the period of power system transformation from a central structure based on coal to a decentralized structure, through an appropriate support mechanism. The government, as a decision-maker, must indicate that the environmental criterion is becoming a priority here.

Further studies will focus on updating the proposed model. The author plans to take into account random phenomena impacting the annual effect of hydrogen production by introducing further decision criteria, including indicators of equipment operation reliability and those of energy supply continuity from the power system.”

  1. in direct connection with point 3 the author does not emphasize the practical implications of this research study. it would be very interesting to see if the main findings of the research would be useful to be implemented in practice. Or maybe if the findings can explain some phenomena within the Poland economic/social/energetic reality

The results of the analyzes can be used to build an appropriate energy policy and organize, for example, development support systems, following the development and support of renewable energy production technologies or cogeneration technologies.

  1. also i don't see the limitations of the research study

Thank you for your comment.

I added this in the research methodology chapter.

The simulations are performed with the following simplification assumptions:

  • quasi-determined state is analyzed, in which the dynamic processes (switching the electrolyzers on and off ) are not taken into account,
  • during the one-year analysis period, random processes that may affect the energy yield, such as interruptions in the electricity supply from the power system (short circuit, power plant failures) are not taken into account,
  • energy supply interruptions from photovoltaic panels due to, e.g. failure of inverters, are not taken into account.

  1. references section must be improved with other significant studies in this scientific area

Thank you for your comment. Literature has been expanded to include the following articles:

    • Siksnelyte, I.; Zavadskas, E.K.; Streimikiene, D.; Sharma, D. An Overview of Multi-Criteria Decision-Making Methods in Dealing with Sustainable Energy Development Issues. Energies 2018, 11(10), 2754
    • Ceran, B.; Kaźmierczak, Ł. The use of Multicriteria Comparative Analysis to the Selection of the Standalone Hybrid Power Generation System Based on Renewable Energy Source. 15th International Conference on the European Energy Market (EEM), 2018, Łódź, Poland, 1-5.
    • Volkarta, K.; Weidmann, N.; Bauer, C.; Hirschberg, S. Multi-criteria decision analysis of energy system transformation pathways: A case study for Switzerland. Energy Policy, 2017, 106, 155-168.
    • Campisi, D., Gitto, S., Morea, D. An Evaluation of Energy and Economic Efficiency in Residential Buildings Sector: A Multi-criteria Analisys on an Italian Case Study. International Journal of Energy Economics and Policy, 2018, 8(3), 185-196.
    • Afgan, N.H.; Veziroglu, A.; Carvalho M.G. Multi-criteria evaluation of hydrogen system options. Int J Hydrogen Energ 2007, 32, 3183-3193.
    • Tapia L.C.F., Vigouroux R.Z.; Silveira J.L. Sustainability Assessment of Hydrogen Production Techniques in Brazil: A Multi-criteria Analysis, Print ISBN: 978-3-319-41614-4, Electronic ISBN: 978-3-319-41616-8
    • Badea, G.; Naghiu, G. S.; Felseghi, R. -A.; Raboaca, S.; Aschilean, I.; Giurca, I. Multi-criteria analysis on how to select solar radiation hydrogen production system. AIP Conference Proceedings, 2015, 1700, 050011.
    • Ren, X.; Shimin W.L.; Donga, D.L. Sustainability assessment and decision making of hydrogen production technologies: A novel two-stage multi-criteria decision making method. Int J Hydrogen Energ 2020, Available online

Round 2

Reviewer 1 Report

The author significantly improved the manuscript according to the suggestions of my previous review report.

Author Response

  Thanks for taking the time to review.

Reviewer 2 Report

The manuscript has been improved. there is some space to further provide enhancement regarding the mathematical model in terms of the normalized parameters. Check again the equations (18)  - (19) and the normalized criterion x'mn. 

Author Response

  Thank you for your comment.   Indeed there was a small markup error, now on line 370 " xnm "   is correctly marked as " x'nm ".       Thank you for your review.

Reviewer 3 Report

Taking into account that the author complied with all my remarks, I have no more suggestions for improvement.

Author Response

  Thank you for your comments and taking the time to review.